# Fecal Microbiota Transplantation in Inflammatory Bowel Disease

**DOI:** 10.3390/biomedicines11041016

**Published:** 2023-03-27

**Authors:** Adrian Boicean, Victoria Birlutiu, Cristian Ichim, Paula Anderco, Sabrina Birsan

**Affiliations:** 1County Clinical Emergency Hospital of Sibiu, 550245 Sibiu, Romania; 2Faculty of Medicine, Lucian Blaga University of Sibiu, 550169 Sibiu, Romania

**Keywords:** microbiota transplantation, inflammatory bowel disease, fecal microbiota transplantation

## Abstract

Inflammatory bowel diseases represent a complex array of diseases of incompletely known etiology that led to gastrointestinal tract chronic inflammation. In inflammatory bowel disease, a promising method of treatment is represented by fecal microbiota transplantation (FMT), FMT has shown its increasing effectiveness and safety in recent years for recurrent CDI; moreover, it showed real clinical benefits in treating SARS-CoV-2 and CDI co-infection. Crohn’s disease and ulcerative colitis are characterized by immune dysregulation, resulting in digestive tract damage caused by immune responses. Most current therapeutic strategies are associated with high costs and many adverse effects by directly targeting the immune response, so modifying the microbial environment by FMT offers an alternative approach that could indirectly influence the host’s immune system in a safe way. Studies outline the endoscopic and clinical improvements in UC and CD in FMT patients versus control groups. This review outlines the multiple benefits of FMT in the case of IBD by improving patients unbalanced gut, therefore improving endoscopic and clinical symptomatology. We aim to emphasize the clinical importance and benefits of FMT in order to prevent flares or complications of IBD and to highlight that further validation is needed for establishing a clinical protocol for FMT in IBD.

## 1. Introduction

Fecal microbiota transplantation (FMT), a therapy based on the microbiome, has acquired widespread interest in the scientific, clinical and lay communities in recent years [1]. Through the use of various techniques and modalities, FMT is a cumbersome procedure that restores a balanced intestinal flora through feces infusion from healthy donors into a diseased gastrointestinal tract (GI) to cure a specific condition. FMT has been successfully used for both non-GI diseases and GI-diseases, the second being represented by idiopathic constipation, recurrent *Clostridioides difficile* infection (CDI), inflammatory bowel disease (IBD) and irritable bowel syndrome [2,3,4,5]. FMT is not a new therapeutic concept, but one that has felt an increasing interest in recent years, with evolving methodology and clinical indications [6]. Many patients may find FMT distasteful and unimaginable, but the concept has been around for decades and the procedure has been proven to be effective [7].

The human gastrointestinal tract contains a wide diversity of microorganisms, of which the bacterial community, made up of at least 10^14^ predominantly anaerobic bacteria, is the dominant one. It should be mentioned that not all species of GI bacteria have been yet cultivated [8]. *Bacteroides*, *Firmicutes phyla, Actinobacteria phyla* and *Fusobacteria* are some of the 1000 species that describe the normal gut microbiota and are genetically imprinted from birth [9]. There are different bacterial communities in the proximal and distal intestine, with most intestinal microbes being hosted by the colon, which plays a critical role in health and illness [10]. The proximal one contributes to the production, absorption and distribution of micronutrients and vitamins; regulation of metabolism and metabolism of xenobiotics; and the renewal of intestinal epithelial cells, but also to the immune system development and protective measures against pathogens. The disruption of the intestinal microbiota, now called dysbiosis, is a state that is believed to be involved in a range of diseases, including irritable bowel syndrome, IBD, enteric infections, colorectal cancer, asthma, atopic diseases, obesity and metabolic syndrome [11]. Genetically susceptible individuals develop IBD when their immune systems respond inappropriately to intestinal microbes. Dysbiosis is also associated with abscesses, surgery at a young age and worsening symptoms in patients with Crohn’s disease (CD) [9].

The severity of IBD and complications have a positive correlation with an overpopulation of pathogenic bacteria (*Coprobacillus*, *Clostridium ramosum*, *Clostridium hathewayi*), and a decrease in the beneficial anti-inflammatory bacteria *Faecalibacterium prausnitzll,* which are beneficial bacteria in the gut that regulate the immune system of the host. Protective bacteria have a role in immunosuppression, preventing induction of cytokines and potential intestinal damage [2,11]. In patients with CD, their microbiota presented a predominance of *Actinomyces* spp., increased *Veillonella* spp. *E. Coli* and *Intestinibacter* spp. The gut microbiota of patients with UC are specified by an increase of *Eubacterium rectum*, *E. Coli* and *Ruminococcus gnavus*, microbes that maintain and induce cell inflammation [2].

In cases of IBD, levels of *Eubacterium rectale*, *Faecalibacterium prausnitzii*, and *Roseburia intestinalis* as well as other healthy bacteria were more reduced in comparison with healthy microbiota from control groups [2].

The intestinal flora composition can be altered by slight, temporary changes due to diet and probiotics or by significant changes due to antibiotics [12]. The majority of risk factors for IBD identified so far are related to the microbiome and include smoking, diet, hygiene hypothesis, exposure to gastroenteritis and early antibiotic use [13]. An important issue is the diet, patients with IBD should receive nutritional recommendations, a diet rich in vegetables and fruits is associated with a decreased risk of developing IBD. On the other hand, ultra-processed food and carboxymethylcellulose predispose patients to IBD development or flares despite proper treatment [12].

The immune system is known to be usually concessive to commensal microbes colonizing in the GI tract. As a potential pathogenic mechanism of IBD, the intestinal microbiota’s imbalance may stimulate an abnormal immune response [14]. Studies with several mutant mouse strains demonstrated the impact of the innate immune system on the intestinal microbiota and the development of gut inflammation, metagenomic analyses of the gut microbiota outlined the co-dependence between intestinal microbes and inflammatory diseases of the gut. Furthermore, studies on mice emphasized that T- and B-cell-deficient mice lacking the T-bet transcription factor developed colitis with an altered microbiota that, upon transfer to wild-type-recipient mice, induced bowel inflammation [13,15,16,17,18,19,20,21,22,23,24,25,26,27,28,29,30]. *Mycobacterium avium* subspecies paratuberculosis and *Fusobacterium varium* are some of the few specific bacteria that have been associated with IBD. However, specific pathogens that fulfill Koch’s postulates have not yet been identified, Koch’s postulates required that the identified organism be present in all cases of the disease, isolated from diseased patients and cause disease when reintroduced to a healthy susceptible animal. However, due to the various triggers involved in CD and UC, Koch’s theory (a pathogen for a disease) is not enough in order to explain the unbalanced immune response of the microbiota in case of IBD [13].

One of the newest and least explored methods of modifying the GI microbiota in IBD involves FMT [8]. In the last decade, FMT has experienced a promising transformation, from being considered an alternative and reserved form of treatment, lacking sufficient medical evidence, to being accepted as a primary effective therapeutic option [6].

We will summarize in this review the latest data and evidence collected regarding the indications and methods by which FMT is used in IBD while comparing the risks and benefits of this procedure, providing a new perspective on the future of this therapeutic potential.

## 2. Material and Methods

Original reports and reviews describing how to use FMT to treat IBD were reviewed using manual and electronic bibliographic sources. English articles were searched in Web of Science, PubMed, Medscape, Up-to-Date and Google Scholar Databases up to December 2022. In this study, 74 publications on FMT’s use in IBD were selected. Below is a list of alternatives used for searching databases: “fecal”, “faecal”, “microbiota”, “microflora”, “feces”, “faeces” and “stool”, individually or with the following options for the transplant lexical field: “transplant”, “transfusion”, “implantation”, “implant”, “donor”, “donation”, “enema”, “reconstitution”, “infusion”, “therapy” and “bacteriotherapy”. In order to ensure a complete search result, all of these terms were searched independently as well as in combination with various IBD descriptive terms such as “Crohn disease”, “Crohn’s disease”, “inflammatory bowel disease”, “colitis”, “ulcerative colitis”, “IBD”, “CD” or “UC”.

## 3. History of FMT

Historically, FMT’s first use in a human patient was reported in the IVth century by Ge Hong in China when it was first administered orally to a patient for food poisoning or severe diarrhea [16]. Li Shizhen described how stool products were used for treating GI symptoms such as constipation, abdominal pain, diarrhea, vomiting and fever during the XVIth century [17]. Fabricius Acquapendente, an Italian anatomist, recounted in the XVIIth century: “I have heard of animals which lost the capacity to ruminate, which, when one puts into their mouth a portion of the materials from the mouth of another ruminant which that animal has already chewed, they immediately start chewing and regain their other health” [18]. FMT then began to be used in veterinary medicine, later being called “transfaunation”. Fecal transplants have been performed for the treatment of horses with diarrhea by administering healthy feces in the rectums of the sick horses. Rumen fluid has also been used to treat cows and alpacas suffering from various gastrointestinal disorders [19].

Eiseman et al. performed and reported the first human fecal transplant in 1958 to treat cases of pseudomembranous colitis. In four patients with refractory infections who failed to respond to antibiotics, fecal retention enemas were found to be effective [20]. A second series of pseudomembranous colitis cases was added in 1981 by Bowden et al., with 16 patients successfully treated with retention enemas as well [21]. Further, a CDI case was treated with FMT for the first time in 1983 and was reported by Schwan et al. [22]. Almost 40 years later, in 2021, microbiota transplantation as a treatment in a large number of CDI patients, recorded very low rates of recurrence [23]. Previous studies on patients with CDI and SARS-CoV-2 co-infection recorded a statistically significant correlation between FMT and the reduction of abdominal pain for the patients at discharge (91.3%, *p* < 0.005) as well as normalization of inflammatory markers: CRP mean 5.67 mg/dL, WBC mean 7695 cells/mL and fibrinogen 420 mg/dL in FMT patients (*p* < 0.05). On the same topic, the study realized by Konturek et al. outlined a significant CRP reduction after FMT therapy in all treated patients. FMT proved its benefits in recurrent CDI, being considered a salvatory option for patients that developed severe and recurrent CDI infection, despite proper antibiotic use [23,24,25,26,27,28,29,30].

Moreover, we aim to outline that very few adverse effects are generally directly attributable to the FMT procedure. Most reported adverse events in the literature have been self-limiting gastrointestinal symptoms comprising abdominal cramps, nausea and constipation. Fever, Gram-negative bacteremia and bowel perforation are very rare adverse effects. Furthermore, in order to improve the safety of FMT, recent studies describe new technics such as washed microbiota preparation, which is based on the use of an automatic microfiltration machine and subsequent repeated centrifugation. In a study with patients who underwent either washed microbiota transplantation (WMT) or crude FMT, in the same FMT center with the same indications, fewer adverse effects were recorded in the WMT group [31].

Up to date studies on FMT outlined its benefits in antibiotic-refractory CDI, autoimmune diseases, behavioral diseases, metabolic disorders and organic diseases. Furthermore, the review realized by Yuanyuan Zhao opened new perspectives, showing progress of fecal microbiota transplantation in liver diseases, through the gut–liver axis [30].

## 4. Inflammatory Bowel Disease

The GI tract is affected by inflammatory bowel disease (IBD), an ongoing chronic recurrent condition that involves chronic remitting and relapsing inflammation [24]. A genetically susceptible individual’s dysregulated immune response to environmental factors is believed to be at the heart of IBD’s etiology [25]. The etiology is still incompletely known, but the dominant hypothesis holds that a pathogenic or altered microbiota in a genetically susceptible individual leads to the inflammation in IBD. Considerable efforts have been made to understand the genetic and immunological basis that determines this disease, and medical treatments are used to remit the patient’s inflammatory response. The medication is represented by amino salicylates, steroids, tumor necrosis factor modulating agents, thiopurines and other immunosuppressants.

The response to treatment is often limited or unsatisfactory due to side effects, infections or even lack of response [11,26]. By understanding the role played by the microbiota in this pathology, new and innovative treatment options can be developed that modulate the gut bacteria [11].

Despite several attempts to identify the bacterial, fungal or viral origins of IBD, as well as the general imbalance of the gut microbiota, the majority of the potential candidates have been rejected due to a lack of valid scientific evidence. Mycobacterium avium subspecies *Paratuberculosis* and *Escherichia Coli* (*E. Coli*) are the specific microorganisms still actively being studied thoroughly [27].

IBD, a chronic recurrent inflammation of the GI tract, is defined by two pathologies: ulcerative colitis (UC), affecting only the colon and its mucosa, or Crohn’s disease (CD), involving any section of the GI tract [11,14]. 

There is an increasing prevalence of UC as a chronic inflammation of the colonic mucosa caused by contact between luminal content and the mucosal immune system. It has a high prevalence of persistent or recurrent symptoms, including anemia with bloody diarrhea and abdominal pain [28,29,30,31,32,33,34,35,36,37,38,39,40,41,42,43,44,45,46,47,48,49,50,51,52,53,54].

CD is a chronic relapsing IBD that has not yet been fully understood, but is thought to be caused by excessive GI immune responses against the microbiota in the gut of genetically susceptible individuals or under an environmental influence [30,31,32,33]. Several studies have indicated that these patients have a dysbiosis of their intestinal microbiota, including an increase in bacteria that can cause inflammation, such as E. Coli, and a reduction in bacteria that prevent inflammation, such as *Faecalibacterium prausnitzii* [30,31,32,33]. Recent clinical trials mention the role of FMT in inflammatory bowel disease (IBD), multiple sclerosis and Parkinson’s disease. This suggests not only a local modulating intestinal effect but also a systemic immunological response to FMT with an impact on the gut–lung, gut–liver and gut–brain axes [25,29,30].

Immunological response after FMT was associated with a substantial reduction in the colonic mucosal CD8+ T cell density and a decrease in serum concentrations of IL-6 and IP-10. Serum levels of IL-6 and VCAM-1 were all significantly correlated with CRP and ESR, as has been highlighted in the study by Yanzhi et. al. on FMT’s role in ulcerative colitis treatment [29]. Additionally, it is considered that FMT has an important role in decreasing gut inflammation via the induction of IL-10 and TGF-β, cytokines critical for T-reg accumulation in the intestine. Moreover, FMT is involved in the inhibition of pathogenic TH-17 cells, through induced IL-10+ T-regs in patients with IBD or Crohn’s disease [29,30,31,32,33,34,35,36,37,38,39,40,41,42,43,44,45,46,47,48,49,50,51,52,53,54,55]. 

## 5. Methods of FMT

FMT has started to be considered an innovative method for treating IBD in severe cases, with a considerable success rate, It is a procedure used to administer a healthy fecal solution into the recipient’s intestinal tract. However, there are a number of questions related to this treatment option that influence the outcome of the procedure, namely: how is the donor selected, what is the procedure of separation of the fecal bacteria, how is the solution preparation made and what is the route of administration [34]? Over time, most successfully treated patients received a single fecal treatment from the donor, although success has also been reported with multiple infusions [35].

FMT composition is still debated, with emphasis on the bacterial content of donated stool, which may come from specific donors or be a combined stool, which provides better results [36]. Initially, patients were allowed to identify their own donors, who were microecologically similar to them, as they were chosen from those with close relations to the patient, close family and friends [14]. Some institutions later offered the option of one or more anonymous donors. As a result, the patient no longer had to identify the donor and a group of tested, healthy donors with a history of healing was created. Additionally, the recipient is spared from interacting with donors who have common genetic or environmental susceptibilities [6]. Stools from chosen donors still need consistent screening to exclude infectious pathogens [37].

The efficacy of combining bacteria from the GI tract from multiple donors is a promising research direction for FMT due to the increased metabolic characteristics and species diversity and colonization [38].

In order to improve the safety for patients undergoing FMT therapy, donors have to fulfill a list of criteria for selection (Table 1). A detailed medical certificate and detailed medical history of the donor must be provided prior to the procedure, as well as stool, serological tests, and most importantly, informed consent [1,2,3,4,5,6,7,8,9,10,11,12,13,14,15,16,17,18,19,20,21,22,23,24,25,26,27,28,29,30,31,32,33,34,35,36,37,38,39,40,41,42].

Fecal analyses focused specifically on bacterial analysis consistently showed that the FMT recipients increased the richness and variety of their microbiota, and that their microbiota characteristics remained similar to those of the donor months and years after the procedure [43,44].

Fresh feces are collected and processed on the day of procedure in most institutions. Stool processing before FMT can also affect grafting. It has been shown that frozen stool at −80 °C conditions can be transplanted as effectively as fresh [11].

According to the protocols for the administration of fresh feces, we must keep in mind that the these must be homogenized and diluted for an easily administration. Using the freshly produced donor stool within 6 h after issuance requires dissolving it in sterile saline. As vehicles, other diluents such as yogurt or milk, can also be used. Larger volumes of prepared solutions tend to yield better results [45,46,47,48,49,50,51,52,53,54,55,56,57,58,59,60,61,62,63,64,65,66,67,68,69,70].

Using highly filtered human microbiota, Hamilton et al. combined it with a cryoprotectant and then froze it at −80 °C for storage until needed. This procedure removed fecal odor and reduced the volume of filtrate [70]. Clinical studies have shown that using standardized, purified feces has the same efficacy as using freshly filtered feces [6].

One of the exclusion criteria in studies was represented by the use of antibiotics 30 days within the intervention of FMT for IBD. In order to achieve successful FMT grafting, patients need to stop their antibiotic treatment at least 24 h before the procedure [39].

Different methods exist for introducing feces for transplantation, including upper GI methods (by nasogastric or gastro-jejunal tube or esophagogastroduodenoscopy), lower GI methods (by retention enema or colonoscopy), oral capsule or even multiple methods for the same recipient [47]. The best route of administration is considered to be dependent on the type and location of IBD. For example, in the case of ulcerative colitis studies show that enema or colonoscopy ensures maximal benefits; in the same vein in the case of CD with ileal inflammation of the gut, gastro-jejunal administration is more effective to reduce the intestinal lesions [27,29,30,33,35]. It is crucial to provide patients with support and education prior to undergoing FMT, regardless of the route of administration or source of the fecal material [47].

FMT in the upper GI tract has several disadvantages, including pain during tube insertion, aspiration risks and no possibility of assessing colonic mucosa or obtaining mucosal tissue samples. By performing FMT via a colonoscopy, the colon is recolonized with beneficial bacteria and cleansed, which reduces the number of residual organisms and spores in the colon [36,48,49]. The retention enema FMT is more preferred by patients due to less invasiveness and high patient acceptability, as it is a less invasive procedure than a colonoscopy. As an alternative, the oral capsule is less invasive and has a high acceptability among patients, but it is more expensive and has a large capsule burden [36,50,51]. In order to improve the safety of FMT, a new approach named washed microbiota transplantation, which uses an automatic microfiltration machine and subsequent repeated centrifugation, seems to present fewer adverse effects compared with crude microbiota [31].

Since SARS-CoV-2 ARN was also identified in feces samples during the pandemic of COVID-19 that appeared at the end of 2019, a new challenge has arisen. The possibility of transmission through sample donation led to an alert that imposed additional restrictions. Many studies and therapeutic treatments of FMT were halted by these requirements, although new algorithms for donor screening have been developed since then [43,52,53]. The end of the year 2022 brought a study that encourages transplantation in patients with CDI and COVID-19 co-infection, with proven beneficial effects such as reducing inflammation, and also the need for antibiotics [54]. Although at the moment according to European Society of Clinical Micro-biology and Infectious Diseases: 2021 update on the treatment guidance document for *Clostridioides difficile* infection in adults FMT is approved only for recurrent or refractory CDI [71], recent studies outline the benefits of this procedure in many other diseases or co-infection. For example, co-infection of CDI and SARS-CoV 2 or CDI and IBD are discussed [30,31,32,33,34,35,36,37,38,39,40,41,42,43,44,45,46,47,48,49,50,51,52,53,54,55,56,57,58,59,60,61,62,63,64,65,66,67,68,69,70,71,72].

## 6. Evolution of IBD Cases Treated with FMT

FMT was first published about for IBD treatment in 1989, in which the author stated he had suffered with UC for 7 years despite receiving sulfasalazine and steroids. After 6 months since the FMT procedure by retention enemas from a healthy donor, he remained asymptomatic [55].

A study conducted in 2006 by Borody and his colleagues found that daily FMT enemas were effective in treating severe UC in 6 adults. They observed clinical improvement earlier than 1 week after the procedure, with complete symptomatic remission for all patients during the first 4 months after the procedure [56].

According to Kunde et al., 10 young adults with mild to moderate UC were treated via FMT in March 2013 by enemas for 5 days daily in the first series of pediatric patients with IBD treated with FMT. At the end of this one-month small pilot study, three of the nine patients were in remission and six of them had a sustained clinical response despite the lack of follow-up [57].

A total of 133 subjects were treated with FMT in a 2014 study (77 with UC, 53 with CD and 3 with indefinite IBD), most of whom were resistant to therapy or dependent on medication. Among all included subjects, 57 (43%) (25 with UC, 31 with CD and 1 with undefined IBD) suffered from a difficult or recurrent CDI. FMT has not been adequately evaluated in clinical trials evaluating its effectiveness in restoring disturbed intestinal microbiota due to poor quality studies and insufficient endpoints and inclusion criteria. From the evaluated data, FMT obtained a reduction of 71% in symptoms of the treated patients. The rate of symptom relief remains constant even when CDI is excluded from our analysis (69%). It should be noted that the exact procedures for FMT in the study were not fully recorded. Nevertheless, most patients were treated with polyethylene glycol lavage or antibiotics that were not specified prior to FMT. A total of 42 patients received FMT via upper routes (nasogastric or naso-jejunal tube and gastroscopy), 20 patients through enema, 23 through colonoscopy and 11 patients had FMT infusions via both upper and lower routes [3].

Another analysis conducted in 2014 analyzed 122 patients with IBD with the note that 3 were excluded from study the because of the FMT enema intolerance. As a result of the cumulative analysis, 119 patients were categorized between mild, moderate and severe. A total of 27 (23%) had mild or mild/moderate disease, 16 (13%) had moderate/severe disease, and 19 (16%) were severe disease patients. Refractory therapy occurred in 10 cases (8%) whereas active disease occurred in 44 cases (37%) and refractory pouching occurred in 5 cases (4%). The clinical remission rate was 45% (54 out of 119). Finally, 12 out of the 16 patients (75%) experienced mucosal healing [58].

A cohort in 2017 that included 30 patients with refractory midgut CD demonstrated a clinical remission of 77% after one month after only one nasoduodenal FMT [59,60].

A group of 555 patients with UC were evaluated in 42 studies reporting on FMT in 2017 (9 case reports, 4 randomized controlled trials, 5 case series and 24 prospective cohort studies out of which 20 were uncontrolled and 4 were controlled). The clinical remission rate was 36% (201 patients out of 555) [61]. In a meta-analysis of 24 cohort studies with 307 individuals, the pooled proportion of individuals with UC achieving clinical remission was moderately heterogeneous (54%) [61].

UC response to FMT was analyzed in 2019 to identify bacterial species and metabolic pathways. In patients who experienced remission, the system contained *Eubacterium*, *Roseburia* and short-chain fatty acids, as well as secondary bile acid biosynthesis, whereas in patients who did not experience remission, the system contained *Fusobacterium*, *Sutterella* and *Escherichia species. Bacteroides* in donor stool were associated with remission in patients receiving FMT, and *Streptococcus* species in donor stool was associated with no response to FMT [62].

It was reported in 2020 that autologous FMT (a-FMT) had similar benefits to heterologous FMT (h-FMT). The purpose of a-FMT is to re-establish the gut microbial community after disturbances have occurred using one’s own feces in a healthy state. IBD and other infectious diseases can be treated with h-FMT in which feces are transplanted into the sick person from a healthy donor. It is preferable to use a-FMT over h-FMT in order to avoid infectious complications; however, it is important to determine which stool samples are functionally optimal in order to prevent complications related to inflammation in IBD (Table 2) [63]. 

Data from the study by Moayyedi and colleagues suggest that the patients newly diagnosed with UC may have better results being treated directly with FMT, suggesting it as an initial treatment possibility. Suffering from intestinal homeostasis could be more easily restored at the beginning of the disease than during its course [24,64].

## 7. Safety of FMT

In order to increase the success rate of FMT in IBD, many factors need to be considered, such as the donor criteria selection, the current disease status and the standardization of the processing protocol [65]. Because there is no long-term information available for this procedure, safety data are limited. FMT has been shown to have a positive effect on patients with IBD in small case series and retrospective studies [39]. The FMT procedure includes potential adverse effects, despite the fact that both patients and doctors believe it to be “natural” or even “organic”. There is a risk of contracting a disease from using the feces of a donor. It seems that adverse events are less frequent in FMT in the lower gastrointestinal tract [66]. Despite being a serious problem, enteric pathogen transmission by FMT seems to be uncommon because of the present donor screening process. Many adverse effects, including moderate fever and mild GI problems, have been associated in IBD patients, especially those with FMT [45]. Adverse reactions include some long-term and some short-term effects (Table 3) [37,38,39,40,41,42,43,44,45,46,47,48,49,50,51,52,53,54,55,56,57,58,59,60,61,62,63,64,65,66,67,68,69,70].

FMT has begun to attract the attention of researchers and more and more studies related to this intervention have appeared. However, in the field of IBD, there are currently far too many unanswered questions, causing some concern among patients and physicians.

In order for transplantation to become an effective treatment for IBD and to be considered a gold standard, the medical world must answer the following questions:Which are the patients with IBD who are expected to have a favorable clinical response post-transplantation? Detailed clinical and paraclinical elements are required.What other types of bacteria, viruses and fungi are involved in this process?Which is the connection (step by step) between the microbiota and IBD?What is the connection between the microbiota, FMT and patients’ medications?What is the connection between the microbiota, FMT and patients’ personal medical histories?Who are the ideal donors and what characteristics do they have?Which is the optimal method of transplantation according to the affected area of the digestive tube?What is the optimal moment for transplantation?How can side effects be reduced?Should FMT or medication be used? A comparison between the safety and effectiveness of the medication used for IBD and the safety and effectiveness of the FMT is needed.What are the parameters that require post-transplant follow-up?

To answer all these questions and improve the safety of the intervention, a multi-center study would be needed to answer them, after which an international protocol would be established. Moreover, partnerships with very advanced laboratories must be made for a detailed sequencing of the microbiota in patients with IBD.

The availability of more clinical data on FMT for IBD will lead to more mechanistic understanding in an area where there has been little information until now. The best way to administer FMT has not yet been demonstrated in clinical practice with good evidence, thus choosing the right approach should be based on the particulars of each case. From the studies reviewed, it would appear that a single fecal transplant would be sufficient for a favorable clinical response (this occurring in about seven days).

The optimal time to perform the transplant remains debatable, and there is no consensus among doctors regarding this aspect. A better efficiency of transplantation has been identified in severe cases of IBD, but transplantation is also encouraged in those newly diagnosed. FMT appears to be both risk-free and successful in preventing recurrent infections in people with IBD.

The results obtained in small-scale studies raise numerous problems. First, there is mistrust about statistical significance, with small studies often obtaining statistically significant but clinically insignificant results. Moreover, there is also very little detail about how the studies were conducted, working methods and the inclusion criteria and exclusion criteria. Moreover, patient selection is very important and must be executed according to well-established criteria.

Currently, there are no studies that identify very long-term patient response. Thus, the lack of data to confirm with certainty the safety and efficiency of the method over a 5-to-10-year interval raises many questions. The possibility of transmission of infectious diseases in the process of microbiota transplantation is a certainty, and the subject requires further investigation. Quite a few specialists have raised the issue of transmission of bacteria/viruses not yet identified or that a routine laboratory cannot detect. This aspect can complicate IBD and mislead the physician.

Exacerbation of IBD can raise serious treatment problems. The question still remains whether the effectiveness of the transplant treatment would help to overcome the acute episode or, on the contrary, would worsen it.

## 8. Conclusions

FMT remains a hope for IBD patients. Although the studies are not extensive enough to lead to the establishment of an international protocol, therapeutic successes can be identified in some cases, which encourages study in this area.

The statistical significance in the studies carried out up to this point is not impressive and the success rates are still low. This is likely due to the fact that not all the details are known regarding the limits of the procedure.

All in all, given the risks that FMT includes and the lack of solid protocols, studies are quite difficult to conduct, and although it is a promising method (and has opened up new opportunities in research), it will take a significant amount of time before this method of treatment will be performed routinely in hospitals around the world.

## Figures and Tables

**Table 1 biomedicines-11-01016-t001:** Donor selection criteria for FMT.

Donor Selection Criteria for FMT	
Absolute Exclusion Criteria	Relative Exclusion Criteria
Failing providing the informed consent	Age under 18 or over 70 years
Systematic and local microbial infections	History of major GI surgery
Irritable bowel syndrome	Neuropsychiatric diseases
Malignant pathologies and chemotherapeutics administration	An unapproved body mass index (smaller or bigger than the normal value of 18–25 kg/m^2^)
Chronic GI disorders	Diabetes mellitus
Peptic ulcer diseases	Systemic autoimmune disease
Administration of antibiotics	Atopic diseases such as asthma and eczema
Immunosuppressive agents and biological agents	Symptoms of chronic ache including chronic fatigue syndrome and fibromyalgia
GI polyposis	Metabolic syndrome
Current communicable diseases	
Increased risky lifestyle due to drug abuse intravenous or dangerous sexual behaviors	
Inflammatory bowel disease	
An autoimmune or atopic illness history or ongoing immunomodulating treatment	
Gastroesophageal reflux disease	
Intestinal microbiota-affecting medications such as prokinetic agents, probiotics, aspirin, proton pump inhibitors or steroids	
Chronic constipation	

**Table 2 biomedicines-11-01016-t002:** Studies of FMT in IBD.

Borody T.J. et al. 2003 [56]	6 patients with severe, recurrent symptoms of UC FMT with enemas for 5 days Complete reversal of symptoms was achieved in all patients by 4 months post-FMT
Kunde S. et al. 2013 [57]	10 pediatric patients with mild-to-severe UC FMT with enemas for 5 days Patients achieved clinical remission at 1 week after FMT
Gianluca Ianiro, et al. 2014 [3]	133 patients (77 with UC, 53 with CD and 3 indefinite BDI) 42 patients received FMT via upper routes (nasogastric or naso-jejunal tube and gastroscopy), 20 patients through enema 23 through colonoscopy 11 patients had FMT infusions via both upper and lower routes for IBD resistant to therapy FMT obtained a reduction of 71% in symptoms
Sudarshan Paramsothy, et al. 2017 [61]	555 patients with UC (42 studies reporting on FMT in 2017) Clinical remission after FMT in 201 cases
Sudarshan Paramsothy, et al. 2019 [62]	81 patients with active UC in a double blind trial FMT for active UC and stool samples from donors FMT group was associated with specific bacteria and metabolic pathways with induction of remission
Samuel P. Costello et al. 2019 [72]	73 adults with UC FMT via colonoscopy followed by 2 enemas over 7 days 38 Donor FMT 35 Autologous FMT
Wang, Y. et al. 2020 [29]	16 active UC patients enrolled 87.5% (*n*= 14) patients achieved clinical response to FMT

**Table 3 biomedicines-11-01016-t003:** Adverse effects of FMT in IBD patients.

FMT in IBD-Adverse Effects	
Short-Term Minor Adverse Effects	Potential Long-Term Adverse Effects
Abdominal tenderness	Infection and/or sepsis (recognized and unrecognized)
Abdominal pain	Obesity
Bloating	Transmission of enteric pathogens
Flatulence	Recurrent IBD
Diarrhea/Constipation	Introducing chronic diseases based on alterations in the intestinal microbiota such as: − colon cancer, − obesity, − atherosclerosis, − diabetes, − asthma, − non-alcoholic fatty liver disease, − autism.
Borborygmus	Disease transmission: cardiometabolic diseases, autoimmune diseases and neurological diseases
Nausea and vomiting especially in patients that have an oral FMT route	Transmitting unrecognized infectious agents that may cause a disease some years later such as HIV or hepatitis C
Transient fever	Abnormally low blood pressure
Adverse effects after sedation	Recurrent UC

## Data Availability

Not applicable.

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
