# Peer review of "Fecal Microbiota Transplantation in Inflammatory Bowel Disease"

_biomedicines, 2023, doi:10.3390/biomedicines11041016_

Round 1
Reviewer 1 Report
Manuscript No Biomedicine-2296180
„Fecal microbiota transplantation in Inflammatory Bowel Disease” for Biomedicine
Comments:
1. In paragraph 2, Authors wrote about Koch's postulates. Please expand on this issue.
2. The title of paragraph 2 is not entirely consistent with the description in the chapter. Please describe the human gastrointestinal microbiota in detail. Please indicate the benefits and potential harms that these microorganisms cause in the body.
3. The history of FMT has been described very superficially. Please indicate what events were groundbreaking, bringing knowledge to the present state. What was significant, causing interest in this issue, etc.
4. Please write the names of bacteria in italics.
5. Authors quite often indicate the active participation of the immune system in the potential activity of FM after transplantation. However, there are no precise descriptions of how the immune response is activated, what cellular components are activated, and what soluble mediators are involved in the reactions. Also, does the reaction always develop as expected? Moreover, whether there may appear adverse reactions?
Author Response
1.In paragraph 2, Authors wrote about Koch's postulates. Please expand on this issue.
We thank the distinguished reviewer for this observation. We modified the paragraph and synthesize Koch's postulates.
” Mycobacterium avium subspecies paratuberculosis and Fusobacterium varium are some of the few specific bacteria that have been associated with IBD. However, specific pathogens that fulfill Koch's postulates have not yet been identified, Koch's postulates required that the identified organism be present in all cases of the disease, isolated from diseased patients, cause disease when reintroduced to a healthy susceptible animal, but due to the various triggers involved in CD and UC, Koch's theory (a pathogen, a desease) is not enough in order to expain the unbalanced immune response of the microbiota in case of IBD. [15]”
2.The title of paragraph 2 is not entirely consistent with the description in the chapter. Please describe the human gastrointestinal microbiota in detail. Please indicate the benefits and potential harms that these microorganisms cause in the body.
We appreciate the distinguished reviewers’ observation in this regard we attached paragraph 2 to the introduction and described in detail the alterations of the gut microbiome in case of CD or UC.
” The severity of IBD disease and complications have a positive correlation with an overpopulation of pathogenic bacteria (Coprobacillus, Clostridium ramosum, Clostridium hathewayi) and a decrease in the beneficial anti-inflammatory bacteria Faecalibacterium prausnitzll, beneficial bacteria in the gut regulates the immune system of the host, protective bacteria have a role in immunosuppression, preventing induction of cytokines and potential intestinal damage [2, 10]. In patients with CD their microbiota presented a predominance of Actinomyces spp., increased Veillonella spp. E, Coli and Intestinibacter spp. The gut microbiota of patients with UC is specific by an increase of Eubacterium rectum, E. Coli and R.gnavus, microbes that maintain and induce cell inflammation. [2]
In cases of IBD levels of Eubacterium rectale, Faecalibacterium prausnitzii, Roseburia intestinalis as well as other healthy bacteria were more reduced in comparison with healthy microbiota. [2]
The intestinal flora composition can be altered by slight, temporary changes due to diet and probiotics or by significant changes due to antibiotics. [12]. The majority of risk factors for IBD identified so far are related to the microbiome and include smoking, diet, hygiene hypothesis, exposure to gastroenteritis and early antibiotic use [15]. An important issue is the diet, patients with IBD should receive nutritional recommendations, a diet rich in vegetables and fruits is associated with a decreased risk of developing IBD, on the other hand ultra-processed food, carboxymethylcellulose predispose patients to IBD development or flares despite proper treatment. [12]”
3.The history of FMT has been described very superficially. Please indicate what events were groundbreaking, bringing knowledge to the present state. What was significant, causing interest in this issue, etc.
We thank the distinguished reviewer for drawing our attention upon this, we included some more data in the paragraph concerning the current state of FMT and some new studies and perspectives of FMT.
”Previous studies on patients with CDI and SARS- COV-2 co-infection recorded a statistically significant correlation between FMT and the reduction of abdominal pain for the patients at discharge (91.3%, p<0.005) as well as normalization of inflammatory markers: CRP 5.67 mg/dl mean, WBC mean 7695 cells/ml, and fibrinogen 420 mg/dl in FMT patients (p <0.05). In the same line the study realized by Konturek et. al. outlined a significant CRP reduction after FMT therapy in all treated patients. FMT proved its benefits in recurrent CDI, being considered a salvatory option for patients that developed severe and recurrent CDI infection, despite proper antibiotic use. [23-30]
Moreover, we aim to outline that very few adverse effects are generally directly attributable to the FMT procedure. Most reported adverse events in the literature have been self-limiting gastrointestinal symptoms comprising abdominal cramps, nausea, and constipation. Fever, gram-negative bacteremia, and bowel perforation are very rare adverse effects. Furthermore, in order to improve the safety of FMT, recent studies describe new technics like washed microbiota preparation, which is based on the use of an automatic microfiltration machine and subsequent repeated centrifugation. In a study with patients who underwent either washed microbiota transplantation (WMT) or crude FMT, in the same FMT center with the same indications, fewer adverse effects were recorded in the WMT group. [31]
Up to date, studies on FMT outlined its benefits in antibiotic-refractory CDI, autoimmune diseases, behavioral diseases, metabolic disorders, and organic diseases furthermore, the review realized by Yuanyuan Zhao opens new perspectives, showing progress of faecal microbiota transplantation in liver diseases, through the gut-liver axis. [30]”
- Please write the names of bacteria in italics.
We thank the distinguished reviewer for this observation, scientific names have been italicized.
5.Authors quite often indicate the active participation of the immune system in the potential activity of FM after transplantation. However, there are no precise descriptions of how the immune response is activated, what cellular components are activated, and what soluble mediators are involved in the reactions. Also, does the reaction always develop as expected? Moreover, whether there may appear adverse reactions?
We thank the distinguished reviewer for raising this important issue and for your provided suggestions in order to improve our manuscript. We have added the following paragraph with regard to the possible immunological response after FMT and we added a table with short term adverse effects and potential long term adverse effects.
” Immunological response after FMT was associated with a substantial reduction in the colonic mucosal CD8+ T cell density and a decrease in serum concentrations of IL-6, and IP-10. Serum levels of IL-6 and VCAM-1 were all significantly correlated with CRP and ESR, as has been highlighted in the study by Yanzhi et. al. on FMT's role in Ulcerative Colitis treatment [29]. Also, it is considered that FMT has an important role in decreasing gut inflammation via the induction of IL-10 and TGF-β, cytokines critical for T-reg accumulation in the intestine. Moreover, FMT is involved in the inhibition of pathogenic TH-17 cells, through induced IL-10+ T-regs in patients with with IBD or Crohn’s disease. [29, 55]”
Reviewer 2 Report
The paper is very interesting and the topic is very actual. The wwork is fit for the publication but I suggest:
-At page 2, line 2, the authors should consider a better and more detailed explanation about the protective and toxic bacteria
- The insertion of a table describing all works about fecal microbiota transplantation in Inflammatory Bowel Disease
Author Response
The paper is very interesting and the topic is very actual. The work is fit for the publication but I suggest:
1.At page 2, line 2, the authors should consider a better and more detailed explanation about the protective and toxic bacteria
We thank the distinguished reviewer for the encouragements and for raising this issue in need of clarification. We have included the following paragraph in order to better describe the protective and toxic bacteria in IBD.
” The severity of IBD disease and complications have a positive correlation with an overpopulation of pathogenic bacteria (Coprobacillus, Clostridium ramosum, Clostridium hathewayi) and a decrease in the beneficial anti-inflammatory bacteria Faecalibacterium prausnitzll, beneficial bacteria in the gut regulates the immune system of the host, protective bacteria have a role in immunosuppression, preventing induction of cytokines and potential intestinal lesions [2, 10]. In patients with CD their microbiota presented a predominance of Actinomyces spp., increased Veillonella spp. E, Coli and Intestinibacter spp. The gut microbiota of patients with UC is specific by an increase of Eubacterium rectum, E. Coli and R.gnavus, microbes that maintain and induce cell inflammation. [2]
In cases of IBD levels of Eubacterium rectale, Faecalibacterium prausnitzii, Roseburia intestinalis as well as other healthy bacteria were more reduced in comparison with healthy microbiota. [2]
The intestinal flora composition can be altered by slight, temporary changes due to diet and probiotics or by significant changes due to antibiotics. [12]. The majority of risk factors for IBD identified so far are related to the microbiome and include smoking, diet, hygiene hypothesis, exposure to gastroenteritis and early antibiotic use [15]. An important issue is the diet, patients with IBD should receive nutritional recommendations, a diet rich in vegetables and fruits is associated with a decreased risk of developing IBD, on the other hand ultra-processed food, carboxymethylcellulose predispose patients to IBD development or flares despite proper treatment. [12]”
2.The insertion of a table describing all works about faecal microbiota transplantation in Inflammatory Bowel Disease
We thank the distinguished reviewer for the appreciation and provided suggestions in order to improve our paper. We included a table describing some of the pilot studies and trials about faecal microbiota transplantation in Inflammatory Bowel Disease (Table no .2)
|
Borody TJ. Et al 2003 |
6 patients with severe, recurrent symptoms of UC FMT with enemas for 5 days. Complete reversal of symptoms was achieved in all patients by 4 months post-FMT.
|
|
Kunde S. Et al 2013 |
10 pediatric patients with mild- to- severe UC. FMT with enemas for 5 days. Patients achieved clinical remission at 1 week after FMT
|
|
Gianluca Ianiro, et. Al 2014 |
133 patients (77 with UC, 53 with CD and 3 indefinite BDI) 42 patients received FMT via upper routes (nasogastric or naso-jejunal tube and gastroscopy), 20 patients through enema 23 through colonoscopy and 11 patients had FMT infusions via both upper and lower routes definite IBD)- IBD resistant to therapy. FMT obtained a reduction of 71% in symptoms |
|
Sudarshan Paramsothy, et. Al ,2017
|
- 555 patients with UC (42 studies reporting on FMT in 2017) -Clinical remission after FMT was in 201 cases.
|
|
Sudarshan Paramsothy et al. 2019
|
81 patients with active UC- double blind trial. -FMT for active UC and stool samples from donors - FMT group was associated with specific bacteria and metabolic pathways with induction of remission.
|
|
Samuel P. Costello et al. 2019 |
-73 adults with UC - FMT via colonoscopy followed by 2 enemas over 7 days - n=38 Donor FMT -n =35 Autologous FMT
|
|
] Wang, Y, et al 2020 |
-16 active UC patients enrolled -87.5%patients achieved clinical response to FMT6 active UC patients
|
Reviewer 3 Report
- - Use oxford comma
- - “Flora”
- Use microbiota or microbiome terms
- - Clostridioides difficile, Faecalibacterium prausnitzii, etc.
Use italics
- - “FMT has been successfully used for both, non-GI diseases and GI-diseases, the second being represented by idiopathic constipation, recurrent Clostridioides difficile infection (CDI), inflammatory bowel disease (IBD) and irritable bowel syndrome”
In addition to recurrent Clostridioides difficile infection, which are the diseases with an approved indication for FMT?
- - “IBD, a chronic recurrent inflammation of the GI tract, is likely to be caused by ulcer-ative colitis (UC), affecting only the colon and it's mucosa or Crohn's disease (CD), being able to involve any section of the GI tract”
IBD is not caused by UC or CD, IBD is UC or CD
- - Define the abbreviations the first time they appear (for example E. coli)
Author Response
We thank the distinguished reviewer for the provided suggestions, we revised the terms and used italics for the bacteria names
-2. FMT has been successfully used for both, non-GI diseases and GI-diseases, the second being represented by idiopathic constipation, recurrent Clostridioides difficile infection (CDI), inflammatory bowel disease (IBD) and irritable bowel syndrome”In addition to recurrent Clostridioides difficile infection, which are the diseases with an approved indication for FMT?
We highly appreciate the distinguished reviewer’s observation in this regard. We added a paragraph with some new studies and benefits of FMT, although at the moment there are protocols only for recurrent or refractory CDI, pilot studies and trials outline the benefits of FMT in IBD.
”Although at the moment according to European Society of Clinical Micro-biology and Infectious Diseases: 2021 update on the treatment guidance document for Clostridioides difficile infection in adults FMT is approved only for recurrent or refractory CDI, [71] recent studies outline the benefits of this procedure in many other disease, or co-infection, for example CDI and SARS-COV 2 or CDI and IBD [30-72]. ”
- - “IBD, a chronic recurrent inflammation of the GI tract, is likely to be caused by ulcerative colitis (UC), affecting only the colon and it's mucosa or Crohn's disease (CD), being able to involve any section of the GI tract” IBD is not caused by UC or CD, IBD is UC or CD.
We thank the distinguished reviewer for raising this important issue, we revised the paragraph.
” IBD, a chronic recurrent inflammation of the GI tract, it is defined by two pathologies ulcerative colitis (UC), affecting only the colon and its mucosa or Crohn's disease (CD), being able to involve any section of the GI tract [10, 13].”
4.- Define the abbreviations the first time they appear (for example E. coli)
We thank the distinguished reviewer for the provided suggestions, we revised the abbreviations.
Reviewer 4 Report
This is a short review covering an interesting topic, I have a few comments that may help improve the quality of the Manuscript:
English language needs editing, also please check abstract. Maybe find a native speaker to proof read the entire Manuscript.
I would like to see more about fecal microbiota transferring in Abstract and less about general IBD... please reconsider
Can more be said about the effectiveness of FMT in other areas, not just IBD?
Can healthy microbiota be described in greater detail? only "pathogens" are mentoined
Page 3, first paragraph is missing refference. please revise
Replace side effects with adverse effects.
Consider italic for names of bacteria.
Does FMT efficacy differ relative to the way it was administered etc? maybe this data can also be included in section 5.
narrative review usually isnt structured with discussion and search strategy, please check. Also, search strategy can be presented in a figure
Author Response
This is a short review covering an interesting topic, I have a few comments that may help improve the quality of the Manuscript:
We thank the distinguished reviewer for the appreciation and provided suggestions. We reconsidered the abstract and described more about FMT, we also added some more information and up-to- date references with regard to the benefits of FMT in other areas like liver pathology, we add the paragraph and reference.
” Recent clinical trials mention the role of FMT in inflammatory bowel disease (IBD), multiple sclerosis, and Parkinson’s disease. This suggests not only a local modulating intestinal effect but also a systemic immunological response to FMT with an impact on gut-lung, gut liver and gut-brain axes [30].”
Zhao, Y.; Gong, C.; Xu, J.; Chen, D.; Yang, B.; Chen, Z.; Wei, L. Research Progress of Fecal Microbiota Transplantation in Liver Diseases. J. Clin. Med. 2023, 12, 1683.
2.Can healthy microbiota be described in greater detail? only "pathogens" are mentioned
We thank the distinguished reviewer for the encouragements and for raising this issue in need of clarification. We have included the following paragraph in order to better describe the protective and toxic bacteria in IBD.
” The severity of IBD disease and complications have a positive correlation with an overpopulation of pathogenic bacteria (Coprobacillus, Clostridium ramosum, Clostridium hathewayi) and a decrease in the beneficial anti-inflammatory bacteria Faecalibacterium prausnitzll, beneficial bacteria in the gut regulates the immune system of the host, protective bacteria have a role in immunosuppression, preventing induction of cytokines and potential intestinal lesions [2, 10]. In patients with CD their microbiota presented a predominance of Actinomyces spp., increased Veillonella spp. E, Coli and Intestinibacter spp. The gut microbiota of patients with UC is specific by an increase of Eubacterium rectum, E. Coli and Ruminococcus gnavus ( R.gnavus), microbes that maintain and induce cell inflammation. [2]
In cases of IBD levels of Eubacterium rectale, Faecalibacterium prausnitzii, Roseburia intestinalis as well as other healthy bacteria were more reduced in comparison with healthy microbiota. [2]
The intestinal flora composition can be altered by slight, temporary changes due to diet and probiotics or by significant changes due to antibiotics. [12]. The majority of risk factors for IBD identified so far are related to the microbiome and include smoking, diet, hygiene hypothesis, exposure to gastroenteritis and early antibiotic use [15]. An important issue is the diet, patients with IBD should receive nutritional recommendations, a diet rich in vegetables and fruits is associated with a decreased risk of developing IBD, on the other hand ultra-processed food, carboxymethylcellulose predispose patients to IBD development or flares despite proper treatment. [12]”
- Page 3, first paragraph is missing reference. please revise
4.Replace side effects with adverse effects.
- Consider italic for names of bacteria.
We thank the distinguished reviewer for this observation, scientific names have been italicized, we also replaced side effects with adverse effects and revised the missing reference.
6.Does FMT efficacy differ relative to the way it was administrated?
We highly appreciate the observation in this regard and we thank again, we revised the paragraph and explained that the efficacy of FMT is related to the type and location of IBD and that studies show that in case of ulcerative colitis the best route in order to ensure maximal benefits are enemas and colonoscopy, on the other hand in case of CD with ileal inflammation gastro-jejunal administration is more effective.
” Different methods exist for introducing feces for transplantation, including: upper GI methods (by nasogastric or gastro-jejunal tube or esophagogastroduodenoscopy), lower GI methods (by retention enema or colonoscopy), oral capsule and even multiple methods for the same recipient [47]. The best route of administration is considered to be dependent on the type and location of IBD, for example in case of ulcerative colitis studies show that enema or colonoscopy ensures maximal benefits, in the same vein in case of CD with ileal inflammation of the gut, gastro-jejunal administration is more effective to reduce the intestinal lesions [27, 29,30,33, 35]. It is crucial to provide patients with support and education prior to undergoing FMT, regardless of the route of administration or source of the fecal material [47- 56].”
We highly appreciate the observation in this regard and we thank again for the provided suggestions, we changed the structure and reconfigured search strategy as material and methods and included the topic from discussion in the main text.
Round 2
Reviewer 1 Report
The manuscript is corrected according to suggestions.